ecology/behaviour/environmental science

*Daption capense*, remote nest cameras, ICESCAPE, SPPYCAMS, Davis Station, Prydz Bay

**Author for correspondence:**
Kimberley Kliska
e-mail: kimkliska@gmail.com

# Phenology-based adjustments improve population estimates of Antarctic breeding seabirds: the case of Cape petrels in East Antarctica

Kimberley Kliska[†], Colin Southwell, Marcus Salton, Richard Williams and Louise Emmerson

Australian Antarctic Division, Kingston, Tasmania, Australia

  KK, 0000-0001-8950-8393

To monitor and conserve a species, it is crucial to understand the size and distribution of populations. For seabirds, population surveys are usually conducted at peak breeding attendance. One of the largest populations of Cape petrels in East Antarctica is at the Vestfold Islands, where environmental and logistical constraints often prevent access to breeding sites at the optimal time for population surveys. In this study, we aim to quantify the contemporary and historical breeding population size of these Cape petrels by adjusting nest counts for variation in breeding phenology using photographs from remote cameras. We also compare spatial distribution between 1970s and 2017/2018. Our results show ground counts occurred outside peak breeding attendance, and adjusting for phenology changed the contemporary and historical population estimates. The Cape petrels showed local intra-island or adjacent-island changes in their distribution between the 1970s and 2017/2018 with no evidence of expanding or restricting their distribution or a significant change in their breeding population size. The results emphasize the importance of accounting for phenology in population counts, where populations are inaccessible at an optimal survey time. We discuss the applications of our research methodology for populations breeding in remote areas and as a baseline for assessing population change.

# 1. Introduction

Seabirds are indicators of change in the environment [1–3], and as terrestrial breeders and marine foragers, they are vulnerable to

[†]Present address: Tasmanian Parks and Wildlife Service, Tasmania.

changes and human impacts in both terrestrial and marine environments [4–6]. Long-term monitoring of the population size and trends of seabirds can inform the effectiveness of conservation management and contribute to our understanding of ecosystem health [5,7]. Indeed, the long lifespan of seabirds means long-term monitoring is essential to distinguish natural variability from human impacts within a population [8]. Estimating the size of seabird populations is commonly achieved by counting the number of breeding pairs or nests [9,10], with temporal observations providing a long-term dataset for understanding population trends and informing conservation management [11,12]. However, surveys of seabirds in remote areas, such as Antarctica, are often constrained by access restrictions and environmental limitations, such as sea ice, for travel and species' phenology and related attendance at breeding colonies dictating optimal survey timing [13]. Variable nest attendance can contribute to availability bias in abundance surveys of seabirds [14], but precise phenology data that can be used to adjust for availability bias is often rarer than population counts in remote areas [15].

Cape petrels have long been recorded breeding along the Antarctic coastline [16–20], yet there are few formal surveys of their populations. Cape petrels are vulnerable to many threats including human disturbance, pollution, overfishing, by-catch and climate change [5,21,22]. Given they feed on Antarctic krill *Euphausia superba* in the Southern Ocean, the Cape petrel is an indicator species in the Commission on the Conservation of Antarctic Marine Living Resources (CCAMLR) Ecosystem Monitoring Program. One of the largest populations of Cape petrels in East Antarctica is at the Vestfold Islands, Prydz Bay. This population was recorded in 1957 when Phillip Law established Davis Station, and prior to our study, the population had been surveyed twice in 1971/1972 and 1974/1975 austral summers. The paucity of scientific study of this population, despite it being close to the year-round occupied Davis Station, is in part due to the early-summer deterioration of fast-ice that prohibits access to islands where the Cape petrels breed, particularly during their incubation and early chick-rearing period. The lack of recent population counts means that understanding how the populations are responding to changes in their environment and accurately quantifying the risk of threats is currently not possible.

While some parts of Antarctica are changing rapidly in the face of climate change, the Prydz Bay region has some relatively stable environmental conditions. Air temperatures and sea ice (duration and cover) in summer across East Antarctica from 1958 to 2013 have not changed significantly, and only a small increase in wind strength has been recorded [23]. Seasonal changes in sea ice duration in the Prydz bay region are more pronounced and complex than elsewhere in East Antarctica, with pockets of significant positive and negative sea ice duration and seasonality trends apparent [24] in the area where this Cape petrel population is expected to forage during summer [25]. In the context of these mixed signals of environmental change in Prydz Bay and East Antarctica, it is useful to assess if the population size and distribution of Cape petrels in the Vestfold Hills has remained stable or changed in recent decades.

In this study, we aim to (i) quantify contemporary breeding population size and map the current distribution of Cape petrels in the Vestfold Hills, East Antarctica and (ii) assess whether breeding population size and distribution have changed over the last four and a half decades. To achieve these aims, we conducted population surveys and quantified nest attendance patterns of the Cape petrels throughout their breeding season to understand how breeding bird phenology could impact population counts. We then adjusted contemporary and historical survey counts of occupied nests for availability bias using the daily nest attendance data. Nest attendance data were obtained from a remote time-lapse camera and used to adjust nest counts following a method previously applied to Adelie penguins (Southwell *et al.* [15]). We also compared distribution maps of occupied nests during contemporary and historical surveys. By using this approach, we highlight how phenology-based adjustments improve the accuracy of population counts of seabirds and provide a baseline of their abundance and distribution for monitoring future impacts to the Cape petrel population in the Vestfold Hills.

# 2. Methods

## 2.1. Study species and region

Cape petrels are medium-sized, black and white mottled flying seabirds that breed on ice-free areas in Antarctica and the sub-Antarctic [26]. They nest in small to large aggregations among sheltered, steep rocky terrain with crevices and ledges on cliffs and slopes [27,28]. Breeding Cape petrels arrive at colonies in Antarctica during October and November and undertake a pre-laying exodus before laying in late November to early December. Arrival date at the breeding site does vary spatially and temporally, but egg laying is more synchronous among years and throughout the range of this species at lower

**Figure 1.** Map of the study area and changes in distribution of occupied nests at the time of the population surveys for Cape petrels *Daption capense* between 1970s and 2017/2018, at the Vestfold Islands, East Antarctica. Island numbers and names as in table 1. Polygons coloured according to inset legend (*b*), where blue represents the presence of occupied nests in both years; yellow where they were found in the 1970s and orange where they were recorded in 2017/2018 only.

latitudes [29,30]. Adults leave in March with chicks fledging soon after [31]. Because the Cape petrel breeding season is in the austral summer and spans across calendar years, we use split-years (e.g. 2017/2018) to recognize that surveys can occur in each calendar year of that breeding season.

Cape petrels forage in the sea ice zone, Antarctic shelf and pelagic waters beyond the shelf waters [25,32] and feed predominantly on krill *Euphausia superba*, with the Antarctic silverfish *Pleuragramma antarcticum* being an addition to the diet during chick rearing [33]. Cape petrels are listed as an indicator species for the CCAMLR Ecosystem Monitoring Program and as 'Stable' and of 'Least concern' on the International Union of Conservation of Nature (IUCN) Red List [34]. The IUCN listing is based on limited information from selected populations.

The study region comprised the Vestfold Hills and nearby offshore islands. The Vestfold Hills is a large area of continental land (45 km long; 30 km wide) in East Antarctica with a northeast to southwest orientated coastline from 68°20′47″ S to 68°40′19″ S. There are 839 islands offshore from the Vestfold Hills, all within 5 km from the continental coastline [27], that are herein referred to as the 'Vestfold Islands' (figure 1). The Vestfold Hills and Islands are mainly ice- and snow-free in the austral summer and provide breeding habitat for six seabird species [20]. To our knowledge, there are no known observations of Cape petrels breeding in the Vestfold Hills, despite over 60 years of expeditioner activity and seabird surveys. The Vestfold Islands are connected to the Vestfold Hills coastline by fast-ice from its formation in autumn until the melt in spring/summer and provides researchers with temporary access to the islands during the early stages of seabird breeding in the area.

Many of the islands in the Vestfold region lack a formally assigned name. To allow accurate and unambiguous identification of un-named Cape petrel breeding locations, we refer to each breeding site using a spatial reference and identification system developed for coastal East Antarctica [35]. This

system provides a unique alpha-numeric identifier, and where present the name, for all islands and coastal outcrops of continental rock across East Antarctica.

## 2.2. Population surveys

Four survey efforts focused on Cape petrel breeding populations have been conducted in the Vestfold Islands in 1971/1972 [27], 1974/1975 (Australian Antarctic Division 1974/1975, unpublished data), 2016/2017 and 2017/2018 (this study) austral summers. We refer to the two early surveys as '1970s' historical surveys because a mix of their data were required to obtain a historical population estimate (explained below), and the two more recent ones as the contemporary surveys. In each of the historical and contemporary surveys, the first year's efforts (i.e. 1971/1972 and 2016/2017) were primarily aimed as reconnaissance surveys to assess the broad distribution of breeding populations, and the second year's efforts (i.e. 1974/1975 and 2017/2018) were aimed to obtained detailed counts at targeted islands where breeding populations were present.

In 1971/1972, the distribution and approximate number of occupied nests were assessed from ground surveys across the Vestfold Islands region. Cape petrels were found only in the southern half of the Vestfold Islands. Colonies were categorized as small (1–30 pairs), medium (30–100 pairs) and large (ca 250 pairs), and the colonies were mapped with low resolution (one symbol per island).

In 1974/1975, all accessible islands in this southern region from Bluff Island south to the Sørsdal Glacier were surveyed for Cape petrels on 17 November and 17 December 1974 from the ground or sea ice (Australian Antarctic Division Davis Biology species log 1974). To ensure consistency of survey dates, both the Davis Station log book 1974/1975 and the personal journal of the biologist who undertook the survey (Richard Williams) were cross-checked for survey dates. In 1974/1975, nest counts and mapping were more precise than in 1971/1972, with the number of occupied nests recorded for smaller regions of each island (e.g. individual slopes) and similarly marked on maps with finer resolution (e.g. slopes/gullies on each island). The 1974/1975 survey captured all islands where Cape petrels were located in the 1971/1972 survey, but nest counts at Turner, Magnetic and Bluff Islands were not conducted due to deteriorating sea ice preventing access: Cape petrels were observed on Bluff Island occupying well-established nests prior to egg laying and their locations mapped with the same resolution (e.g. slopes/gullies). The islands north of Magnetic Island (herein referred to as the 'Northern Islands'; figure 1) were also opportunistically searched during seal surveys conducted from 1 to 8 November 1974, and no sign of breeding Cape petrels was recorded (Williams 2020, personal communication); this was after birds have returned to breeding sites and prior to the pre-laying exodus [30].

In 2016/2017, an on-ground presence–absence survey was conducted on the 19 November 2016 in the southern region to determine whether Cape petrels were still breeding in the area and to assess feasibility for making ground or photographic counts of nesting sites from the sea ice. While not exhaustive across every island, the survey indicated that Cape petrels were still breeding at most sites reported in the 1970s. The presence of Cape petrels was also noted during aerial surveys across the Vestfold Islands between 30 November and 3 December. While observations from the air may fail to detect small populations, they were very likely to detect medium to large aggregations; there were no Cape petrels observed in the northern region during the 2016/2017 aerial surveys.

The 2017/2018 survey effort focused on all the islands where breeding Cape petrels had been previously observed breeding and was conducted over three days (18, 20 and 30 November 2017). At each breeding colony, a combination of ground searches and/or binocular counts was conducted from a vantage point on the sea ice tens of meters perpendicular away from Cape petrel breeding areas. Similar to 1974/1975, the number of occupied nests was recorded for smaller regions of each island and marked on maps with finer resolution (e.g. count per slope, gully or distinguishable terrain feature). As for the 1974/1975 survey, the Northern Islands were opportunistically searched for Cape petrels during comprehensive seal surveys (ground and aerial) between 5 and 13 December 2017, and no Cape petrels were observed.

In the years between the historical and recent surveys, there has been frequent and extensive expeditioner activity in this area, including various ground-based and aerial surveys, and to our knowledge no Cape petrels have been observed in the Northern Islands. We are therefore confident that counts presented in this study encompass the entire Vestfold Islands population of Cape petrels.

## 2.3. Daily nest attendance counts

The ideal time to count populations of breeding seabirds is during early incubation, after all birds have laid eggs and before there are many failures [36]. Importantly, Cape petrels, like many other seabirds, are

known to undertake a pre-laying exodus [37] early in the breeding season before egg lay, and the optimal time for population surveys is shortly after the birds have returned to their colonies. However, in the Vestfold Islands, the timing of the Cape petrel breeding population surveys was dictated by access to islands over fast-ice and could not be optimally timed around breeding phenology (dates presented above under Population surveys).

To adjust sub-optimally timed survey counts to a standardized metric—the maximum number of occupied nests—we obtained date-specific adjustment factors derived from a daily number of occupied nests of Cape petrels during the breeding season using images recorded by an automated time-lapse camera [38,39]. This camera was established overlooking part of the Cape petrel breeding colony at Bluff Island (figure 1) in the Vestfold Islands at 68°33′15.84″ S, 77°54′19.8″ E from 5 November 2019 until the end of the breeding season (and beyond). The camera captured 10 images per day (1 h apart, through the middle of the day) and included 27 Cape petrel nests within its field of view. Camera images were processed using manual image processing software, SPPYCAMS v. 1.0.1 [40], to obtain the number of occupied nests in the camera field of view on the first, middle and last image of each day. To account for variation in the three images counted per day (i.e. first/middle/last images), particularly outside the incubation period, the maximum count on each day was taken to represent the number of occupied nests on that day and used as a basis to estimate date-specific adjustment factors (methods outlined below). Active breeding nests were identified during the early egg laying period (3–7 December) and were nests where adults were observed sitting tight over a few days. The adults then occupying each of these nests were counted throughout the breeding season. Partner birds or birds not on a nest were not counted. These counts ($c$) were then standardized against the maximum number of occupied nests during the breeding season (i.e. $c/27$) as a basis for estimating date-specific adjustment factors (methods outlined below).

## 2.4. Contemporary and historical count data

To quantify contemporary and historical breeding population size of Cape petrels in the Vestfold Islands, we analysed count data from the 2017/2018 and 1970s breeding seasons. To fill the data gaps at three islands without survey counts in 1974/1975 (Magnetic, Turner and Bluff Islands), we used the counts from the 1971/1972 survey; the count data from other islands in 1971/1972 were not used in the historical population estimate because of their lower resolution (using count categories and single icons per island for mapping). Similarly, the 2016/2017 survey was used as a basis to develop count methodology and confirm broad presence–absence prior to 2017/2018 survey effort, but did not contribute to the contemporary population estimate.

To account for potential 'count uncertainty' in the survey nest counts (e.g. guano was mistakenly recorded as an occupied nest, or some breeding nests were hidden from view), we assumed the survey nest counts were within ±10% (with 95% confidence) of the true number present, following Woehler et al. [41].

## 2.5. Adjusting survey data for variable nest attendance

Methods to adjust for availability bias in population counts (e.g. associated with variable nest attendance) have been developed for and applied to Adelie penguin survey data [38]. In this study, we applied this method to flying seabirds. We adjusted the Cape petrel survey count data to estimate the maximum number of occupied nests established in the incubation period by applying the parametric bootstrap implementation in the software ICESCAPE [42]. The process requires reference data of attendance counts through time (e.g. daily nest attendance from camera images), counts of observed abundance (e.g. survey nest counts), the dates of counts and the certainty of counts [42]. Given these data, the adjustment process involved the following steps:

  (i) Deriving a normal distribution of plausible observed abundances using the original count value as a mean and the assumed 10% count uncertainty value as a 95% confidence interval.
  (ii) Modelling the standardized camera time series counts using a generalized additive model (GAM) and the penalized regression spline implementation of Wood [43] in which the degree of smoothness is automatically selected by minimizing a generalized cross-validation (GCV) criterion. Overfitting, which is occasionally observed when using GCV, was controlled by increasing the smoothing parameter ($\gamma$) to 1.4 as recommended by Kim & Gu [44] and Wood [43].

(iii) Generating a distribution of adjustment factors by repeated sampling from the fitted GAM. The sampling was repeated across the fitted GAM's confidence interval at the date of the count. If only a date range, rather than a specific date, was known for a count, samples were also drawn across the date range. This was the case for the 1971/1972 data which contributed historical nest counts for Turner, Magnetic and Bluff Islands. In these cases, we made a conservative assumption that the nest counts were completed between 5 December 1971 (during egg laying and when sea ice access is sometimes possible) and 30 January 1972 (when chicks are present and boat access is possible) and samples were drawn across this range with equal probability.

(iv) Calculating 1000 bootstrapped replicate estimates of the maximum number of occupied nests by randomly drawing values from the count and adjustment factor distributions and iteratively calculating the product of the counts and the inverse of the adjustment factors.

(v) Summarizing the distribution of adjusted population estimates by the median and 95% confidence interval, where a 100 (1 − α)% confidence interval was taken as the α/2 and 1 − α/2 percentile points.

## 2.6. Temporal population change

To obtain contemporary and historical breeding population estimates of Cape petrels in the Vestfold Islands in each of the two analysed survey periods (2017/2018 and 1970s), the 1000 island-specific bootstrap replicate estimates from ICESCAPE for each year were summed across the 17 occupied islands to create a regional distribution of population estimates for each year, which were each then summarized with a regional median and 95% confidence interval. To test whether the regional population sizes in 2017/2018 and 1970s differed, we examined the direction (sign), median and 95% confidence limits of the 1000 differences (2017-minus-1970s) in paired bootstrap replicate estimates. We concluded that the 2017 population was larger than the 1970s population if the median of 2017-minus-1970s bootstrap differences was positive and the 95% confidence limits did not include zero, smaller if the median difference was negative and the 95% confidence limits did not include zero, and that there was no detectable difference between years if the 95% confidence interval of differences included zero.

To investigate changes in the spatial distribution of Cape petrel breeding populations in the Vestfold Islands between contemporary and historical surveys, discrete breeding colonies were identified on each island. A 'breeding colony' was defined as a discrete group of nests with breeding birds in close proximity to one another. In many cases, colonies were separated by environmental features, such as gullies, spur lines, aspect and ridges. For each survey, the location of a breeding colony was transposed from hand-drawn maps onto digitized maps. We then drew a line around the outer perimeter of breeding colonies to represent the maximum extent, using R v. 4.1.1 [45]. This allowed for a coarse comparison of presence and absence of known breeding colonies on islands between the 1970s and 2017/2018. This distribution change could not be examined with finer resolution because site location from historical data did not include GPS, and hand-drawn maps were difficult to interpret at a finer resolution.

# 3. Results

## 3.1. Nest attendance and nest counts

Cape petrels were present on Bluff Island (in the Vestfold Islands) from 5 November 2019 when the camera was installed and, with the exception of a pre-laying exodus from 24 to 26 November when no adults were present, maintained a presence through to 8 March 2020. The pattern of nest attendance showed an initial increase in mid-November (to approximately half the number during early incubation), then there was a drop during the pre-laying exodus through to early egg laying (2–7 December) with all nests occupied on 20 December (figure 2). Nest attendance fluctuated slightly until late January, when fewer adults occupied their nests and chicks were left unattended. From our observations, the optimal time to survey these breeding Cape petrels based on their phenology is at the start of December when incubation is at its peak.

After adjusting Cape petrel population counts for attendance, the estimated maximum number of occupied nests in the Vestfold Islands in 2017/2018 was 1845 (1681–2163) (table 1). This was based on a survey which identified 763 occupied nests (table 1). In 2017/2018, Cape petrels were distributed

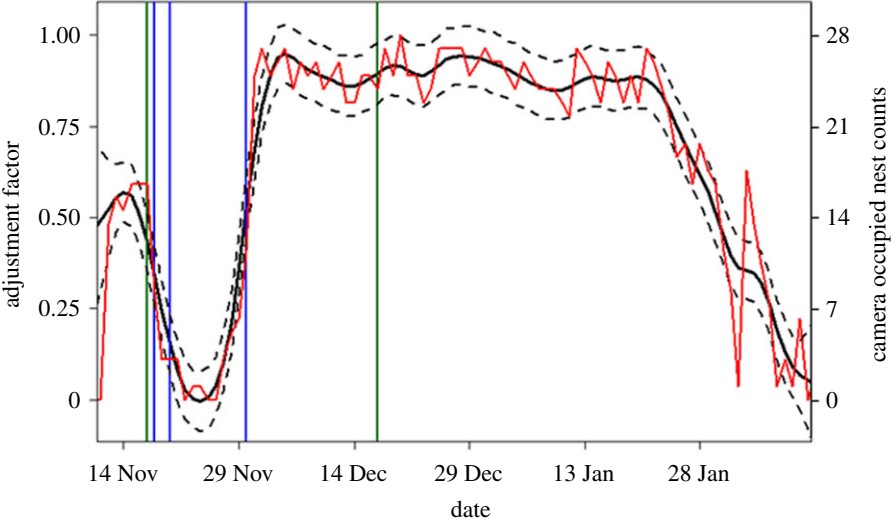

**Figure 2.** GAM adjustment factor from ICESCAPE (solid black line) with the 95% confidence intervals (dashed black lines, left *y*-axis). The camera occupied nest counts (red line, right *y*-axis) are from time-lapse camera images of Cape petrels at the Bluff Island in 2019/2020. Vertical lines indicate the timing of surveys in the 1970s (green) and 2017/2018 (blue).

**Table 1.** Counts of Cape petrel occupied nests in the southern area of the Vestfold Islands at the time of surveys in the 1970s and 2017/2018, and the estimated maximum number of occupied nests after adjusting for nest attendance at the time of surveys. Values are medians and 95% confidence intervals. Zero indicates the island was searched and no occupied nests were recorded. Table breaks align with the three panels in figure 1.

| Island number (name) | 1970s | 2017/2018 | 1970s adjusted data | 2017/2018 adjusted data |
|---|---|---|---|---|
| 72276 (Gardner) | 15 (14–16) | 0 | 17 (6–27) | 0 |
| 72270 (Bluff) | 330 (297–362) | 406 (366–446) | 763 (635–940) | 767 (646–926) |
| 72260 (Magnetic) | 50 (46–54) | 23 (21–25) | 58 (48–64) | 43 |
| 72266 (Turner) | 50 (46–54) | 122 (110–133) | 58 (48–64) | 230 (194–277) |
| 72390 (Mule) | 46 (41–50) | 3 (3–3) | 51 (45–59) | 20 (13–38) |
| 72406 | 7 (6–8) | 0 | 8 (7–9) | 0 |
| 72351 (Hawker) | 10 (9–11) | 1 (1–1) | 11 (10–13) | 12 (4–12) |
| 72450 | 0 | 15 (13–16) | 0 | 97 (64–194) |
| 72519 | 2 (2–2) | 0 | 5 (4–6) | 0 |
| 72509 | 9 (8–10) | 0 | 21 (17–26) | 0 |
| 72461 (Kazak) | 271 (246–299) | 30 (27–33) | 303 (266–344) | 195 (128–402) |
| 72419 (Zolotov) | 295 (265–323) | 97 (87–107) | 330 (284–374) | 287 (226–384) |
| 72438 | 30 (27–33) | 10 (9–11) | 33 (29–39) | 29 (23–40) |
| 72486 | 44 (39–48) | 4 (4–4) | 49 (43–56) | 12 (9–16) |
| 72503 | 38 (34–42) | 11 (10–12) | 42 (37–49) | 32 (25–44) |
| 72522 | 55 (49–60) | 10 (9–11) | 62 (54–70) | 30 (23–39) |
| 72536 (Pintado) | 115 (105–126) | 31 (28–34) | 129 (112–145) | 91 (71–121) |
| totals | 1367 (1315–1422) | 763 (719–805) | 1940 (1797–2133) | 1845 (1681–2163) |

over 13 of the 17 islands, and the maximum number of occupied nests (median values, after adjustment) ranged across islands from 12 to 767.

The maximum number of occupied nests for the 1970s was estimated to be 1940 (1797–2133), from a survey count of 1367 occupied nests (table 1). In the 1970s, Cape petrels were distributed over 13 of the 17

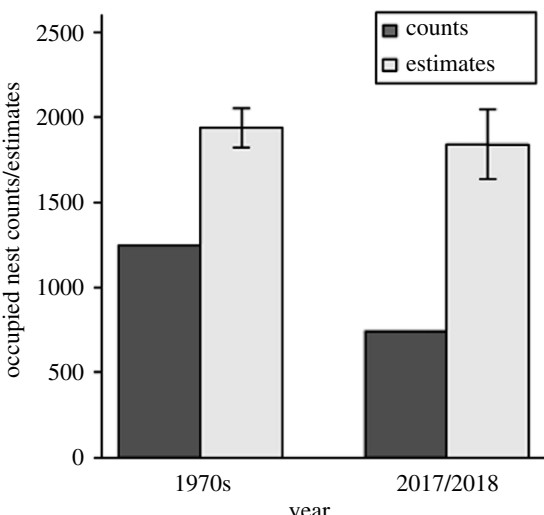

**Figure 3.** Comparison of survey counts (black) and adjusted estimates (grey, with 95% confidence intervals) of occupied nests for Cape petrels at the Vestfold Islands in the 1970s and 2017/2018.

islands, and the maximum number of occupied nests (median values, after adjustment) ranged across islands from 5 to 763.

The adjusted nest counts for the Cape petrel population in the Vestfold Islands are similar for the 1970s to 2017/2018 (figure 3) and the 95% confidence interval of differences included zero, indicating there was no detectable difference between them.

## 3.2. Distribution of occupied nests

Our results show no large-scale change in the distribution of Cape petrel occupied nests in the Vestfold Islands between the 1970s and 2017/2018, with only local intra-island or adjacent-island shifts in distribution (figure 1). Cape petrels were not recorded breeding north of Magnetic Island in either survey period or during opportunistic observations between surveys. Four islands (Gardner Island, IS_72406, IS_72509 and IS_72519) had nesting sites in the 1970s, with between 8 and 21 occupied nests, but no occupied nests were found on these islands in 2017/2018. Some unconfirmed nests were recorded on three of these islands (IS_72406, IS_72509 and IS_72519) during the 2017/2018 survey. Due to the timing of the 2017/2018 survey of those islands aligning closely with the Cape petrel exodus period, it is uncertain whether these islands still had active nesting sites or if they were simply absent at the time of the survey. In 2017/2018, a nesting site was observed on IS_72450 (between Zolotov and Kazak), where they were not detected in the 1970s. The observed 15 occupied nests at this island in 2017/2018 equates to 97 occupied nests after adjustment for nest attendance at the time of the 2017/2018 survey. Within an island, some nesting sites that were observed in the 1970s had no evidence of breeding birds (e.g. nests, or guano-stained rocks) in 2017/2018. Conversely, on some islands nesting sites were detected in some areas in 2017/2018 where they had not been located in the 1970s (figure 1).

## 4. Discussion

Our study demonstrates the importance of adjusting breeding population survey counts for phenology patterns of seabirds. We demonstrate that this approach, applicable to a broad range of seabird species, has improved a contemporary population estimate for one of the largest populations of Cape petrels in East Antarctica. Furthermore, by applying this approach to historical survey data, we suggest that this population has been stable for the last 40 years, with minimal change in distribution. Importantly, if the survey counts were not adjusted for the Cape petrel phenology, the population estimates would have suggested a decrease in the population over the last four decades.

Seabird phenology has implications for population monitoring and understanding how seabirds respond to environmental change [13,15,46]. For the first time, our study reports the daily attendance of Cape petrels at a breeding colony in East Antarctica using data from an automated time-lapse

camera. Automated cameras have been used to monitor wildlife populations globally, and recently in Antarctica to monitor Adelie penguins [13,15,47]. Here, we provide proof of concept using a minimal disturbance monitoring method to successfully use cameras to record breeding phenology of surface-nesting, flying seabirds. This method detected the timing of the pre-laying exodus, which was closely aligned with the timing of survey occupied nest counts. Without adjusting survey data to account for variation in nest attendance, a population decline may have been concluded, whereas adjusted nest estimates showed in fact there had been no change, highlighting the importance of applying this methodology to data with inconsistent survey timing. These adjustments may be particularly important for species with large changes in nest attendance patterns throughout the breeding season, and particularly useful to correct surveys of populations in remote areas, where it is challenging to access breeding sites at the ideal time to census breeding populations.

The similarity of population estimates in the 1970s and 2017/2018 suggests there has been little change in the Cape petrel breeding population size over this period of time in the Vestfold Islands. This aligns with stability in at-sea observations of seabirds, including Cape petrels, in the Prydz Bay region between 1980 and 2001 [48]. Similarly, the Cape petrel population at Ardery Island, near Casey Station in the Windmill Islands, was relatively stable between the mid-1980s and mid-1990s [18]. Collectively, these findings support the current conservation status 'Stable' and 'Least Concern' of Cape petrels breeding across East Antarctica as per the IUCN Red List of Threatened Species [34], although greater certainty of this would be provided through additional contemporary surveys. However, not all Cape petrel populations in Antarctica are apparently stable. At other locations, such as the Antarctic Peninsula, breeding populations of Cape petrels appear to be decreasing [16,22]. At Hop Island, approximately 20 km south of the Vestfold Islands, breeding pairs of Cape petrels appear to have increased by more than 300% from 1986 to 1996 [28]. Investigating population changes can be complex. The observed increase at Hop Island occurred at a local island level. Substantial local-scale population changes were also observed at some sites in our study despite no large change at a population or regional level, suggesting these local changes can be absorbed into the broader population, due, for example, to movement between breeding sites. Notably, the timing of surveys was not always reported, making it unclear whether the survey results from other locations need adjusting for breeding phenology as highlighted by this study.

Spatial distribution of breeding seabirds can be influenced by a range of factors, including suitable habitat, population dynamics, nest fidelity, environmental change and carrying capacity [49]. Cape petrels nest in low sheltered areas [28], and their occupation of nesting areas may be linked with shelter from prevailing winds and storms, and possible absence of snow cover [29]. Annual variation in weather, such as precipitation and storm events, could influence the distribution of nests on an island or between adjacent islands among years. In this case, the minimal movement of Cape petrel nesting sites does align with the relatively stable environment within this region [23]. However, with westerly wind forecast to strengthen [50], more intra-island shifts may be seen in the future as Cape petrels respond to environmental change. More frequent observations of nesting sites or of individually marked birds would assist in assessing site fidelity of Cape petrels between breeding seasons, and this is one consideration for future monitoring given that changes in sea ice conditions, climate change and other human pressures are forecast to impact flying seabirds [12,50].

While our methodology can improve population estimates, there are some limitations of our study due to knowledge gaps in understanding seabird phenology in Antarctica. There are few studies into the breeding phenology of Cape petrels and they are focused on populations in the northern extent of the Antarctic Peninsula [29,37]. Therefore, we are unsure if breeding phenology between the historical surveys in the 1970s and our recent surveys has remained consistent. Further refinements to the approach we have taken here can be achieved by improving our understanding of the temporal (between years) variation in phenology. In this case, the phenology data (i.e. nest attendance) are assumed to apply across the Vestfold Islands and to the historical and contemporary surveys. The Vestfold Islands area is relatively small (approx. 20 km of coastline), and therefore assuming limited spatial variation across this area is reasonable. In previous applications of this phenology-based adjustment method to contemporary and historical Adélie penguin population counts [15,38], it was possible to account for spatial and temporal variation in attendance by deriving adjustment data from a network of multiple cameras and across multiple years. In this first application to a flying seabird species, we drew on a single year of nest camera images to determine nest attendance counts in line the recent nest count surveys. Further improvement in applying this approach to seabird population estimates will be possible as more cameras are deployed at flying seabird colonies and the amount of nest attendance data grows over years.

In addition to these limitations, other survey methodology such as aerial photography at an appropriate separation distance to minimize the chance of wildlife disturbance [51,52] may further improve our ability to monitor Cape petrel populations in this and other areas. However, Cape petrels have a brown and white mottled wing pattern that may affect detectability if using aerial photography in rocky areas. This may also be a limitation to observers conducting ground counts as per our study. If aerial photography methods are to be used in the future we recommend investigating if mottled colouring will impact on the results.

Wildlife in Antarctica, including Cape petrels as a flying seabird, are likely to be impacted by environmental change and/or increased human activity in the future. Our ability to monitor, assess and mitigate any such impacts is paramount in achieving the objectives of the Antarctic Treaty for avoiding detrimental changes in the distribution, abundance or productivity of species or populations of fauna and flora [53]. Our study provides a methodology which includes ground-based surveys and camera images to account for breeding phenology to reliably estimate and monitor Cape petrel population size in an area where environmental and logistical conditions present challenges for optimally timing surveys. Our contemporary estimate of the Cape petrel breeding population in the Vestfold Islands provides a baseline for assessing future population change. This research is timely given the expected focus for National Antarctic programmes, including Australia, to either establish, expand and/or modernize their operations in Antarctica [54]. We recommend the phenology-based adjustment method presented here be used for species where surveying at an optimal time is unachievable due to logistical or environmental constraints.

Ethics. The protocols and procedures for deploying the remote cameras used to obtain data for this study were approved by the Australian Antarctic Division Animal Ethics Committee through Australian Antarctic Science Study projects nos. 4087 and 4518.

Data accessibility. The data and code to reproduce this study can be found online at the Australian Antarctic Division Data Centre at https://data.aad.gov.au/metadata/records/AAS_4518_Cape_Petrel_population_estimates_VestfoldIslands (doi:10.26179/bxja-2451).

Authors' contributions. K.K.: conceptualization, data curation, formal analysis, investigation, methodology, project administration, resources, visualization, writing—original draft and writing—review and editing; C.S.: conceptualization, data curation, formal analysis, funding acquisition, investigation, methodology, project administration, resources, supervision, validation, writing—original draft and writing—review and editing; M.S.: conceptualization, data curation, formal analysis, investigation, methodology, project administration, resources, validation, visualization, writing—original draft and writing—review and editing; R.W.: data curation, methodology, resources, validation, writing—original draft, writing—review and editing; L.E.: conceptualization, formal analysis, funding acquisition, investigation, methodology, project administration, resources, supervision, writing—original draft and writing—review and editing.

All authors gave final approval for publication and agreed to be held accountable for the work performed therein.

Competing interests. We declare we have no competing interests.

Funding. We received no funding for this study.

Acknowledgements. We thank the Australian Antarctic Division and Davis Station personnel for logistics support for fieldwork during the 1970s and recent years. Recent survey approaches were approved through the Australian Antarctic Division Animal Ethics Committee. This work contributes to the objectives of and received support through Australian Antarctic Science projects 4088 and 4518. We would like to acknowledge the contributions from three anonymous reviewers who provided constructive comments to improve this manuscript.

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
