## [Peer Review File · Royal Society Open Science]

Review History

RSOS-210052.R0 (Original submission)

Review form: Reviewer 1

Is the manuscript scientifically sound in its present form?

No

Are the interpretations and conclusions justified by the results?

Yes

Is the language acceptable?

Yes

Do you have any ethical concerns with this paper?

No

Have you any concerns about statistical analyses in this paper?

Yes

Recommendation?

Major revision is needed (please make suggestions in comments)

Comments to the Author(s)

Please see attached document (Appendix A).

Review form: Reviewer 2**Is the manuscript scientifically sound in its present form?**

No

Are the interpretations and conclusions justified by the results?

No

Is the language acceptable?

Yes

Do you have any ethical concerns with this paper?

No

Have you any concerns about statistical analyses in this paper?

No

Recommendation?

Reject

Comments to the Author(s)

This study aims at showing that one Cape petrel population in East Antarctica has been stable in the last ca. 50 years. Authors used survey data adjusted for seasonal changes in the number of breeders. While the study raises an interesting and important issue, I think it fails to convincingly show that the Cape petrel population has been stable. The study is based on more or less two surveys (1974 and 2017) and uses data from another season (2019) to make some phenological adjustment. The conclusions assume that the phenological changes in the number of breeders during the 2019 season are the same as in 1974 and 2017. This is a very strong assumption and authors do not provide any evidence that this is the case. Consequently, there is no evidence that the adjusted counts given by the authors in this study are better than the raw counts and the trend in this petrel population cannot be assessed with certainty.

Additional comments:

Line 35-36: just a detail but photographic ground surveys and surveys from remotely operated cameras are also ground counts. Consider changing "ground count" line 35 with "direct count" or equivalent.

Line 48-51: there are no national territories in Antarctica, only claims. You should modify this section as such statements are not necessary in a scientific paper and only open the door for criticism. You should only mention East Antarctic and the role of CCAMLR without getting into such administrative/political considerations.

Line 58: you could specify Antarctic krill *Euphausia superba* here. Also, should it be "the Cape petrel is an indicator species..." rather than "cape petrels are an indicator species..."?

Line 85: the SAM is not a local environmental parameter and not specific to the Prydz Bay. This is a large climatic mode not relevant here to discuss local environmental changes.

Lines 87-88: mentioning changes in sea-ice here while you just said that the environment, including sea-ice was stable, is a bit strange?

Lines 83-95: this paragraph is a bit blurry. You mention a stable environment but then some environmental changes and then put the Adelie penguin into the story... I don't think this is needed into your story to argue that the environment has been stable or not (at least not at this stage in the introduction) so I would suggest to remove this paragraph. If you decide to keep it, you should present some more solid arguments to conclude that the foraging habitat of the Cape petrel has been stable since 1958.

Methods:

Population surveys: this is not so clear to me why 2016 is not included in the analyses? The 2016 survey is mentioned line 137-139 but then does not appear anywhere? 1972 is barely used as well... overall, I find it hard to follow the survey descriptions and what has been used or not...

Timing, spatial coverage, methods.... seem to vary among survey years so that any comparison between these numbers are based on very soft grounds, whatever the adjustments you make.

Line 183-185: where does this come from? Is this based on any data or is this just a guess?

Lines 198-201: then you assume that there is no change in this egg laying dynamics among years?... this is a very strong, and likely unrealistic, assumption

Line 206-207: this is really misleading to call "repeatability" the 10% arbitrary threshold used to define potential variation around the count estimate. Repeatability has a specific definition which does not correspond to yours here.

Discussion

Line 315: hard to conclude with 2 datapoints...there could have been a lot of changes in this period...

Line 337-338: high site fidelity and stability are two different things. You may have a stable population even if site fidelity is low and vice versa. This section is mixing both concepts.

Line 368: "changes in sea-ice conditions" instead, not "sea ice conditions" per se.

Line 399-400: but this is doable for any kind of organism. The key is not to do it, but to show whether or not it gives accurate numbers. Your study has used this "adjustment" method and made conclusions based on it but the validity of this method is still unknown for your system.

Decision letter (RSOS-210052.R0)

Dear Mrs Kliska

The Editors assigned to your paper RSOS-210052 "Phenology based adjustments to population survey data show no temporal change in the status or distribution of Cape petrels in the Vestfold Islands." have made a decision based on their reading of the paper and any comments received from reviewers.

Regrettably, in view of the reports received, the manuscript has been rejected in its current form. However, a new manuscript may be submitted which takes into consideration these comments.

We invite you to respond to the comments supplied below and prepare a resubmission of your manuscript. Below the referees' and Editors' comments (where applicable) we provide additional requirements. We provide guidance below to help you prepare your revision.

Please note that resubmitting your manuscript does not guarantee eventual acceptance, and we do not generally allow multiple rounds of revision and resubmission, so we urge you to make every effort to fully address all of the comments at this stage. If deemed necessary by the Editors, your manuscript will be sent back to one or more of the original reviewers for assessment. If the original reviewers are not available, we may invite new reviewers.

Please resubmit your revised manuscript and required files (see below) no later than 26-Oct-2021. Note: the ScholarOne system will 'lock' if resubmission is attempted on or after this deadline. If you do not think you will be able to meet this deadline, please contact the editorial office immediately.

Please note article processing charges apply to papers accepted for publication in Royal Society Open Science (<https://royalsocietypublishing.org/rsos/charges>). Charges will also apply to papers transferred to the journal from other Royal Society Publishing journals, as well as papers submitted as part of our collaboration with the Royal Society of Chemistry (<https://royalsocietypublishing.org/rsos/chemistry>). Fee waivers are available but must be requested when you submit your manuscript (<https://royalsocietypublishing.org/rsos/waivers>).

Thank you for submitting your manuscript to Royal Society Open Science and we look forward to receiving your resubmission. If you have any questions at all, please do not hesitate to get in touch.

on behalf of Dr Denise Greig (Associate Editor) and Pete Smith (Subject Editor)
openscience@royalsociety.org

Associate Editor Comments to Author (Dr Denise Greig):

Associate Editor: 1

Comments to the Author:

This is an interesting study and it is great idea to use remote photography to generate a phenology based correction factor for petrel nest counts. Both reviewers wondered whether it was scientifically sound to apply the nesting phenology documented over one year to other years without further validation of the concept; similarly, I was curious whether nesting phenology at 1 island (out of 13) could be extrapolated to the other islands. I don't know if there are good references out there to back up these assumptions, or if you plan to do additional years to ground truth this methodology? It would be good to note in the manuscript that this correction factor could be further refined with data from additional years. And if phenology does shift over time or among locations, it will be difficult to document shifts in population numbers.

Both reviewers offered detailed suggestions for improving the focus and clarity of the manuscript and I hope you will re-submit once you address their concerns.

Reviewer comments to Author:

Reviewer: 1

Comments to the Author(s)

Please see attached document

Reviewer: 2

Comments to the Author(s)

This study aims at showing that one Cape petrel population in East Antarctica has been stable in the last ca. 50 years. Authors used survey data adjusted for seasonal changes in the number of breeders. While the study raises an interesting and important issue, I think it fails to convincingly show that the Cape petrel population has been stable. The study is based on more or less two surveys (1974 and 2017) and uses data from another season (2019) to make some phenological adjustment. The conclusions assume that the phenological changes in the number of breeders during the 2019 season are the same as in 1974 and 2017. This is a very strong assumption and authors do not provide any evidence that this is the case. Consequently, there is no evidence that the adjusted counts given by the authors in this study are better than the raw counts and the trend in this petrel population cannot be assessed with certainty.

Additional comments:

Line 35-36: just a detail but photographic ground surveys and surveys from remotely operated cameras are also ground counts. Consider changing "ground count" line 35 with "direct count" or equivalent.

Line 48-51: there are no national territories in Antarctica, only claims. You should modify this section as such statements are not necessary in a scientific paper and only open the door for criticism. You should only mention East Antarctic and the role of CCAMLR without getting into such administrative/political considerations.

Line 58: you could specify Antarctic krill *Euphausia superba* here. Also, should it be "the Cape petrel is an indicator species..." rather than "cape petrels are an indicator species..."?

Line 85: the SAM is not a local environmental parameter and not specific to the Prydz Bay. This is a large climatic mode not relevant here to discuss local environmental changes.

Lines 87-88: mentioning changes in sea-ice here while you just said that the environment, including sea-ice was stable, is a bit strange?

Lines 83-95: this paragraph is a bit blurry. You mention a stable environment but then some environmental changes and then put the Adelie penguin into the story... I don't think this is needed into your story to argue that the environment has been stable or not (at least not at this stage in the introduction) so I would suggest to remove this paragraph. If you decide to keep it, you should present some more solid arguments to conclude that the foraging habitat of the Cape petrel has been stable since 1958.

Methods:

Population surveys: this is not so clear to me why 2016 is not included in the analyses? The 2016 survey is mentioned line 137-139 but then does not appear anywhere? 1972 is barely used as well... overall, I find it hard to follow the survey descriptions and what has been used or not... Timing, spatial coverage, methods.... seem to vary among survey years so that any comparison between these numbers are based on very soft grounds, whatever the adjustments you make.

Line 183-185: where does this come from? Is this based on any data or is this just a guess?

Lines 198-201: then you assume that there is no change in this egg laying dynamics among years?... this is a very strong, and likely unrealistic, assumption

Line 206-207: this is really misleading to call "repeatability" the 10% arbitrary threshold used to define potential variation around the count estimate. Repeatability has a specific definition which does not correspond to yours here.

Discussion

Line 315: hard to conclude with 2 datapoints...there could have been a lot of changes in this period...

Line 337-338: high site fidelity and stability are two different things. You may have a stable population even if site fidelity is low and vice versa. This section is mixing both concepts.

Line 368: "changes in sea-ice conditions" instead, not "sea ice conditions" per se.

Line 399-400: but this is doable for any kind of organism. The key is not to do it, but to show whether or not it gives accurate numbers. Your study has used this "adjustment" method and made conclusions based on it but the validity of this method is still unknown for your system.

===PREPARING YOUR MANUSCRIPT===

===PREPARING YOUR REVISION IN SCHOLARONE===

Author's Response to Decision Letter for (RSOS-210052.R0)

See Appendix B.

RSOS-211659.R0

Review form: Reviewer 1

Is the manuscript scientifically sound in its present form?

Yes

Are the interpretations and conclusions justified by the results?

Yes

Is the language acceptable?

Yes

Do you have any ethical concerns with this paper?

No

Have you any concerns about statistical analyses in this paper?

No

Recommendation?

Accept with minor revision (please list in comments)

Comments to the Author(s)

The authors did a great job improving the manuscript and I believe they addressed all the comments pointed out by the previous review. I have just a few minor comments (mostly typos):

Abstract:

Line 13: add a comma after "In this study"

Line 14: I know that it can be complicated due to the usual word limitation of the abstract, but I think it would be important to refer that you used photographs obtained from a remote camera to adjust the nest counts

Introduction:

It is better structured now and the aims are well defined.

Line 37: I would add commas before and after "such as Antarctica"

Line 38: I would add commas before "such as sea ice"

Line 54: It would be better to include references based on Cape petrels studies and not only include a review of seabird threats around the world and from different seabird species. Some references could be:

Favero, M., Khatchikian, C.E., Arias, A., Silva Rodriguez, M.P., Cañete, G. and Mariano-Jelicich, R. (2003). Estimates of seabird by-catch along the Patagonian Shelf by Argentine longline fishing vessels, 1999–211106111061. *Bird Conservation International*. 13(4): 273–281.

Braun, C., Esefeld, J., Savelieva, L., & Peter, H. U. (2021). Population decline of the cape petrel (*Daption capense*) on King George Island, South Shetland Islands, Antarctica. *Polar Biology*, 44(9), 1795–1801.

Methods:

Line 94: something is missing after the scientific name of the silverfish, maybe add "being" before "an addition..."

Line 96: Add "of Nature" after Conservation

Line 143: Could the petrels be on pre-laying exodus during this survey?

Line 242: I believe this is point 5

Line 244: Put the title of the subsection in a new line

Results:

Line 274: Figure 2 does not seem to reach the 8th of March

Line 280-282: This is not a result, I suggest you move it to the discussion

Line 288-291: I believe it will be better to describe first the historical data and then the contemporaneous, but this is just a suggestion

Line 290: I believe you meant 16 and not 13 islands, based on table 1

Line 295 – 296: This is not a result, I suggest you move it to the discussion

Discussion

Line 350: add a comma after “Here”

Line 352: it would be good if you point out somewhere in the discussion that this methodology is useful for surface breeders, as many other flying seabirds breed in burrows and your methodology probably would not be feasible or at least not easily applicable

Line 389: Please put Cape with a capital letter.

Line 411: I don't understand what you mean by the body of nest attendance data

Review form: Reviewer 3

Is the manuscript scientifically sound in its present form?

Yes

Are the interpretations and conclusions justified by the results?

Yes

Is the language acceptable?

Yes

Do you have any ethical concerns with this paper?

No

Have you any concerns about statistical analyses in this paper?

No

Recommendation?

Accept as is

Comments to the Author(s)

Based on the comments from the previous reviews, I think the authors have done a good job in improving the manuscript. I found the methodology clearly set out. However, it is still a limitation that you do not ground truth your results, and you claim (now in your title) that this approach improves estimates its surely weakened by this fact. You apply the same approach to two years and assume that the results you get are more accurate.

While some specific aims of the paper are set out in the abstract and end of the introduction, the overall aim of the paper is still not that clear. Is it a proof of concept paper? Or a study to determine changes in the population trend of Cape petrels? One sentence on this should be explicitly written in the abstract and the end of the introduction.

If this is proof of the use of combining phenological data with survey data to estimate counts then I was surprised by the lack of review of literature in the introduction, on existing methodological approaches to do so. Would be nice for the reader to know how developed this is as an approach.

Despite these points I do think this is an interesting and valuable paper, highlighting an approach which could be highly valuable for survey species in challenging environments. I found it well written and structured, especially after the comments from previous reviewers.

Some very minor comments:

l. 53-56 would move these sentences about threats to petrels a bit earlier

l 88 - is this an appropriate reference?

fig 1 change order in label so its yellow (1970s) blue (1970s and 2017/18) and then orange (2017/18)

116-119 why not historical vs contemporary?

l. 264 'R framework?' More conventional to refer to a package

Decision letter (RSOS-211659.R0)

Dear Mrs Kliska

On behalf of the Editors, we are pleased to inform you that your Manuscript RSOS-211659 "Phenology-based adjustments improve population estimates of Antarctic breeding seabirds: the case of Cape petrels in East Antarctica" has been accepted for publication in Royal Society Open Science subject to minor revision in accordance with the referees' reports. Please find the referees' comments along with any feedback from the Editors below my signature.

Please submit your revised manuscript and required files (see below) no later than 7 days from today's (ie 07-Mar-2022) date. Note: the ScholarOne system will 'lock' if submission of the revision is attempted 7 or more days after the deadline. If you do not think you will be able to meet this deadline please contact the editorial office immediately.

on behalf of Dr Denise Greig (Associate Editor) and Pete Smith (Subject Editor)
 openscience@royalsociety.org

Associate Editor Comments to Author (Dr Denise Greig):

Thank you so much for your thoughtful revision and response to the reviewers. I think the manuscript is in good shape. There are just a few additional requests from the reviewers and three typos that I noted:

Line 115. Change "focussed" to "focused"

Line 243. Is there supposed to be a "." after "100"?

Line 244. Move "Temporal population change" to its own line.

Reviewer comments to Author:

Reviewer: 1

Comments to the Author(s)

The authors did a great job improving the manuscript and I believe they addressed all the comments pointed out by the previous review. I have just a few minor comments (mostly typos):

Abstract:

Line 13: add a comma after "In this study"

Line 14: I know that it can be complicated due to the usual word limitation of the abstract, but I think it would be important to refer that you used photographs obtained from a remote camera to adjust the nest counts

Introduction:

It is better structured now and the aims are well defined.

Line 37: I would add commas before and after "such as Antarctica"

Line 38: I would add commas before "such as sea ice"

Line 54: It would be better to include references based on Cape petrels studies and not only include a review of seabird threats around the world and from different seabird species. Some references could be:

Favero, M., Khatchikian, C.E., Arias, A., Silva Rodriguez, M.P., Cañete, G. and Mariano-Jelicich, R. (2003). Estimates of seabird by-catch along the Patagonian Shelf by Argentine longline fishing vessels, 1999–211106111061. *Bird Conservation International*. 13(4): 273–281.

Braun, C., Esefeld, J., Savelieva, L., & Peter, H. U. (2021). Population decline of the cape petrel (*Daption capense*) on King George Island, South Shetland Islands, Antarctica. *Polar Biology*, 44(9), 1795–1801.

Methods:

Line 94: something is missing after the scientific name of the silverfish, maybe add "being" before "an addition..."

Line 96: Add "of Nature" after Conservation

Line 143: Could the petrels be on pre-laying exodus during this survey?

Line 242: I believe this is point 5

Line 244: Put the title of the subsection in a new line

Results:

Line 274: Figure 2 does not seem to reach the 8th of March

Line 280-282: This is not a result, I suggest you move it to the discussion

Line 288-291: I believe it will be better to describe first the historical data and then the contemporaneous, but this is just a suggestion

Line 290: I believe you meant 16 and not 13 islands, based on table 1

Line 295 – 296: This is not a result, I suggest you move it to the discussion

Discussion

Line 350: add a comma after “Here”

Line 352: it would be good if you point out somewhere in the discussion that this methodology is useful for surface breeders, as many other flying seabirds breed in burrows and your methodology probably would not be feasible or at least not easily applicable

Line 389: Please put Cape with a capital letter.

Line 411: I don't understand what you mean by the body of nest attendance data

Reviewer: 3

Comments to the Author(s)

Based on the comments from the previous reviews, I think the authors have done a good job in improving the manuscript. I found the methodology clearly set out. However, it is still a limitation that you do not ground truth your results, and you claim (now in your title) that this approach improves estimates its surely weakened by this fact. You apply the same approach to two years and assume that the results you get are more accurate.

While some specific aims of the paper are set out in the abstract and end of the introduction, the overall aim of the paper is still not that clear. Is it a proof of concept paper? Or a study to determine changes in the population trend of Cape petrels? One sentence on this should be explicitly written in the abstract and the end of the introduction.

If this is proof of the use of combining phenological data with survey data to estimate counts then I was surprised by the lack of review of literature in the introduction, on existing methodological approaches to do so. Would be nice for the reader to know how developed this is as an approach.

Despite these points I do think this is an interesting and valuable paper, highlighting an approach which could be highly valuable for survey species in challenging environments. I found it well written and structured, especially after the comments from previous reviewers.

Some very minor comments:

l. 53-56 would move these sentences about threats to petrels a bit earlier

l 88 – is this an appropriate reference?

fig 1 change order in label so its yellow (1970s) blue (1970s and 2017/18) and then orange (2017/18)

116-119 why not historical vs contemporary?

l. 264 ‘R framework?’ More conventional to refer to a package

===PREPARING YOUR MANUSCRIPT===

one version should clearly identify all the changes that have been made (for instance, in coloured highlight, in bold text, or tracked changes);

===PREPARING YOUR REVISION IN SCHOLARONE===

- If you are requesting a discretionary waiver for the article processing charge, the waiver form must be included at this step.
- If you are providing image files for potential cover images, please upload these at this step, and inform the editorial office you have done so. You must hold the copyright to any image provided.
- A copy of your point-by-point response to referees and Editors. This will expedite the preparation of your proof.

- Ensure that your data access statement meets the requirements at <https://royalsociety.org/journals/authors/author-guidelines/#data>. You should ensure that you cite the dataset in your reference list. If you have deposited data etc in the Dryad repository, please only include the 'For publication' link at this stage. You should remove the 'For review' link.
- If you are requesting an article processing charge waiver, you must select the relevant waiver option (if requesting a discretionary waiver, the form should have been uploaded, see 'File upload' above).
- If you have uploaded any electronic supplementary (ESM) files, please ensure you follow the guidance at <https://royalsociety.org/journals/authors/author-guidelines/#supplementary-material> to include a suitable title and informative caption. An example of appropriate titling and captioning may be found at https://figshare.com/articles/Table_S2_from_Is_there_a_trade-off_between_peak_performance_and_performance_breadth_across_temperatures_for_aerobic_scope_in_teleost_fishes_/3843624.

Author's Response to Decision Letter for (RSOS-211659.R0)

See Appendix C.

Decision letter (RSOS-211659.R1)

Dear Mrs Kliska,

I am pleased to inform you that your manuscript entitled "Phenology-based adjustments improve population estimates of Antarctic breeding seabirds: the case of Cape petrels in East Antarctica" is now accepted for publication in Royal Society Open Science.

Please remember to make any data sets or code libraries 'live' prior to publication, and update any links as needed when you receive a proof to check - for instance, from a private 'for review'

URL to a publicly accessible 'for publication' URL. It is good practice to also add data sets, code and other digital materials to your reference list.

on behalf of Dr Denise Greig (Associate Editor) and Pete Smith (Subject Editor)
openscience@royalsociety.org

Appendix A

The authors compile the historical information of the breeding distribution and number of nests of Cape petrels in Vestfold Islands and compare it with contemporaneous information collected by the authors mainly in 2017/2018. As the surveys were performed on different dates throughout the breeding period, the authors adjusted the number of nests found based on the variation of the number of occupied nests inferred from photographs obtained from a remote camera deployed in one of the studied colonies in 2019. The idea of correcting the estimation of breeding pairs based on the variation of the number of occupied nests throughout the breeding period is very interesting and can be very useful to species that breed in remote areas difficult to access. Nevertheless, the article needs to be improved:

- It is unclear to me which are the main objectives of the article.
- The methodology used needs to be better explained.
- Both the abstract and the introduction need to be restructured.

Please see below my comments and suggestions on how to improve the article.

TITLE

The title does not seem adequate as the adjustments were applied to the number of nests counted, not to the distribution of the species (in which you observed intra-island changes). By status, I assume you mean population trend, and with only 2 years of data that you finally used, it may be too early to draw conclusions about the population trends. Maybe you can change it to population size. I would recommend you something like:

"Phenology based adjustments to population survey data show no temporal change in nest number of Cape petrels..."

Or even change it to:

The importance of using phenology based adjustments to correct the population size of seabirds breeding in difficult to access areas: the case of Cape petrels

If you want to include the distribution in the title, I recommend you take into account the adjusted number of nests in the distribution map and respectively discussion, i.e., instead of only mapping the location of breeding areas you could also represent the number of nests estimated for each of the areas if you have this information by breeding area.

However, if your main objective is to show the general lack of changes in the population trend and distribution of the species, then you could change the title to:

Contemporaneous and historical (similar) population size and distribution of Cape petrel breeding in the Vestfold Islands, East Antarctica.

ABSTRACT

Although I understand the word limits of an abstract, I would recommend some restructuration of the abstract:

For example, begin to explain that it is crucial to know the distribution and population size of a species for conservation and monitorization purposes. Then, explain that some species breed in areas where researchers can only access under specific environmental conditions which may not match with the optimal period to perform population surveys as the number of breeding pairs may change throughout the breeding season. This is the example of Cape petrel breeding in Vestfold Islands in East Antarctica where sea and ice condition limit population surveys. Subsequently explain your main aim and what you have done to achieve it. Delete the sentence from line 14 to 16 as I believe it is not needed in the

abstract. Organize the description and discussion of your results and then present a general conclusion of your article. Right now you have for example in lines 18-20 a sentence explaining your results but the previous sentence sounds more like a conclusion and so it should be at end of the abstract

KEYWORDS:

The common name of the study species is already in the title and the abstract, instead, you could put its scientific name. Also consider changing the "East Antarctica" keyword that is already in the title and abstract, with other keywords that are also relevant but not included in the abstract, for example, some of the software you use, or remote camera

INTRODUCTION:

I would also recommend some re-structuring of the introduction. But before any of that, for me, it seems that the objectives of this article are not well defined, although in the title you gave the main relevance to the use of the phenology adjustment approach, at end of the introduction you don't even mention it. So for you, what are your main objectives in this article? Think about it and restructure the introduction accordingly to that.

Right now you begin to explain the importance of seabirds as indicators of environmental changes and the difficulties of monitoring them. Afterwards, you refer to some of the methods used to estimate seabird populations (although part of the approaches is mainly referred to those species who nest at ground surface), the difficulty to access remote areas and the need to take into account the variation of the number of active nests throughout the breeding period. You already finished the second paragraph explaining that in this article you used the phenology data from a remote photo-trap camera to apply to your data without explaining which species, why that species, etc. You should present this information later on in the introduction

In the third paragraph, you describe several conservation management details about the Antarctic territory but are this needed in the introduction to understand the objective of this article? If you considered that is relevant leave it like it is, otherwise consider putting this information in the methodology section. You finish this third paragraph with an aim of this article however usually the objectives are presented at the end of the introduction and as it is written here it does not have much sense. What is that you want to inform about CEMP? The population estimation? The population changes from 1974 to 2017? Delete this sentence.

In the previous paragraph, you explained why you choose Cape petrels as your study model and here in the fourth paragraph, you describe the species. Although the line of thoughts if correct, some of the information that you put in this paragraph is not relevant for the introduction or its relevance need to be better explained. For example, why is it important their main type of prey? Or the colour of their plumage? If you explain that the colour of their plumage may preclude the use of some methodologies, or that the type of prey is what makes this species a good indicator of the marine environment, then this information would be useful, but right now it does not seem relevant and I recommend you put in the methods.

In the paragraph that begins in line 83, you talk about the environmental characteristics of Antarctica. Consider integrating this information into the third paragraph that you already talk about Antarctica. It is not clear why you mention here the Adele penguin nor why you expect Cape petrels population to be stable instead of increase as in Adele penguins and if there is any study that shows how the wind may influence the breeding distribution of this species or a related species please include it here.

Line 29: add " about the effectiveness of" after the word "inform"

Lines 61-70: Most of the information in this paragraph should go to methods. Why is it relevant to know their prey in the introduction?

Line 97: What do you mean by population status? If you are referring to the population trend, what is the difference between objective 1 and 2? Because for you to understand the trend of a population, you need to compare the changes between the population size in at least two different periods and this is what you are already referring to in objective 2.

METHODS:

Consider to include a "study species" section to put some of the information you included in the introduction but that would fit better here.

The population surveys section is confusing maybe you could organize it per year, i.e., in one paragraph you may explain everything regarding the year 1972 and 1974. Another one to 2017, etc.

Line 125: It is not clear how the data of 2016 was used

Lines 127-128: I would include here the detailed breeding phenology that you put in lines 64-66 of the introduction.

Line 130: one example is enough, but if you want to put all, include also the one of 2016/17

Line 147: Three? why are you excluding 2016? Is it because only presence-absence was register and not the number of nests?

Lines 14-151: And for those islands that were surveyed in both years did you use only the data of 1974, or did you calculate the mean number of nests for both years?

Line 170: I don't understand why you are referring to this here "were considered consistent with the breeding pairs estimates in historic surveys". Isn't this one of the things you want to verify in this article?

Line 184: How was this 10% calculated?

Lines 194-196: But if you don't know when the survey took place how are you going to adjust the survey counts?

Lines 201-203: It is not necessary to put the sign "-" as you already included S, but you have to put E in the longitude. Include also the dates that the camera stayed in the field collecting data. Since you adjust the number of nests based on the pictures of this single-camera I think you need to describe better all the procedures related to it. How did you program the camera? I mean how many pictures per day? How did you choose the place to deploy it? Did you make any population survey to the area covered by the camera to validate the number of nests estimated by the SPPYCAMS software?

Line 204: I am assuming that in reference 38 it's explaining, but how much is the efficiency of this software to correctly identify the number of occupied nests? And does this software distinguish between breeders and non-breeders (floater or loafing individuals that you refer to in line 172) that also visit the colony?

Line 206: How did you calculate this repeatability?

Line 234: why is 2016 not included here if you only take into account presence-absence?

Line 236-238: If possible, specify which areas were visited and on which dates.

RESULTS:

Lines 242-243: rephrase maybe to "...November, when no adults were observed"

Line 247: Cite figure 2 here. Is this species synchronous? I think this is an important detail because if your species is not synchronous, its phenology may change depending on the environmental conditions and you would be introducing a bias in your estimations.

Line 256: How did you test it?

Line 261: 5 to 763

Table 1: To be consistent in the adjusted data columns you should put always "No data" if the number of nests counted was zero as you did for the Northern island or change in Northern islands from "no data" to 0

Table 1: Why did you put ranges in line 11 y 14, but not in line 8 or 10 of 2017? In lines 8 and 10 were the nests only counted once?

Table 1 (line 21): What does this asterisk * mean?

Figure 1: maybe you could represent the estimated number of nest per year in the maps, in this way it would be possible to observe geographic changes in the number of nests

Figure 2: Which family did you use in the gam model? And which response variables did you include? Is this the output of the software ICESCAPE?

Figure 3: This information is already in the last line of table 1, so I am not sure which is the value of this figure?

DISCUSSION

Lines 353-354: You refer that environmental conditions such as snow accumulation influence the breeding success of the species. Were the environmental conditions between 1974, 2017 and 2019 similar? You should check it and if they were not the same you should discuss possible biases. Furthermore, I would recommend you to continue with the remote camera in the field for several years (if that is possible) to understand how the environmental conditions may affect the number of active breeding pairs and potential variation in the breeding phenology of the species.

Line 399: Not sure why this would implicate a novelty. Both species breed on the ground independently of being walking or flying species

Line 401: Drones have some advantages but also some limitation. Usually, you can only operate in specific environmental conditions (low wind speed), the surveys are limited by the duration of the battery, it may scare the birds which may fly away, there can be a limitation regarding the maximum distance between the operator and the drone and this may limit its use depending on the environmental conditions. Satellite images have also been used to count penguins, boobies or albatrosses nests, but not sure if this would be feasible with Cape Petrels due to a lack of contrast between the bird's plumage and the ground and because of their small body size.

Appendix B

Response to reviewers:

Associate Editor (Dr Denise Greig) Comments:

This is an interesting study and it is great idea to use remote photography to generate a phenology-based correction factor for petrel nest counts. Both reviewers wondered whether it was scientifically sound to apply the nesting phenology documented over one year to other years without further validation of the concept; similarly, I was curious whether nesting phenology at 1 island (out of 13) could be extrapolated to the other islands. I don't know if there are good references out there to back up these assumptions, or if you plan to do additional years to ground truth this methodology? It would be good to note in the manuscript that this correction factor could be further refined with data from additional years. And if phenology does shift over time or among locations, it will be difficult to document shifts in population numbers.

Both reviewers offered detailed suggestions for improving the focus and clarity of the manuscript and I hope you will re-submit once you address their concerns.

Reply: In line with these comments and those of the two reviewers, we have revised the ms to emphasise the primary objective is to estimate the current population size of cape petrels, and to do this we use a phenology-based adjustment. We have rephrased terms to clarify we are not assessing a trend over time, but rather make a comparison with earlier surveys in 1971/72 and 1974/75.

In addition, we included a paragraph in the discussion to highlight the uncertainty associated with spatial/temporal variability in phenology.

“While our methodology can improve population estimates, there are some limitations of our study due to knowledge gaps in understanding seabird phenology in Antarctica. There are few studies into the breeding phenology of Cape petrels and they are focused on populations in the northern extent of the Antarctic Peninsula [27, 35]. Therefore, we are unsure if breeding phenology between the historical surveys in the early 1970s and our recent surveys has remained consistent. Further refinements to the approach we have taken here can be achieved by improving our understanding of the temporal (between years) variation in phenology. In this case, the phenology data (i.e. nest attendance) are assumed to apply across the Vestfold Islands and to the historical and contemporary surveys. The Vestfold Islands area is relatively small (~20 km of coastline), and therefore assuming limited spatial variation across this area is reasonable. In previous applications of this phenology-based adjustment method to contemporary and historical Adélie penguin population counts [15, 36], it was possible to account for spatial and temporal variation in attendance by deriving adjustment data from a network of multiple cameras and across multiple years. In this first application to a flying seabird species we drew on a single year of nest camera images to determine nest attendance counts in line the recent nest count surveys. Further improvement in applying this approach to seabird population estimates will be possible as more cameras are deployed at flying seabird colonies and the body of nest attendance data grows over years.”

Reviewer 1 comments:

The authors compile the historical information of the breeding distribution and number of nests of Cape petrels in Vestfold Islands and compare it with contemporaneous information collected by the authors mainly in 2017/2018. As the surveys were performed on different dates throughout the

breeding period, the authors adjusted the number of nests found based on the variation of the number of occupied nests inferred from photographs obtained from a remote camera deployed in one of the studied colonies in 2019. The idea of correcting the estimation of breeding pairs based on the variation of the number of occupied nests throughout the breeding period is very interesting and can be very useful to species that breed in remote areas difficult to access. Nevertheless, the article needs to be improved: -

It is unclear to me which are the main objectives of the article. –

The methodology used needs to be better explained. –

Both the abstract and the introduction need to be restructured. Please see below my comments and suggestions on how to improve the article.

Reply: We have restructured the abstract and introduction, clarified the objectives and elaborated on aspects of the methodology following the reviewer's suggestions detailed below

TITLE

The title does not seem adequate as the adjustments were applied to the number of nests counted, not to the distribution of the species (in which you observed intra-island changes). By status, I assume you mean population trend, and with only 2 years of data that you finally used, it may be too early to draw conclusions about the population trends. Maybe you can change it to population size. I would recommend you something like: "Phenology based adjustments to population survey data show no temporal change in nest number of Cape petrels..." Or even change it to: The importance of using phenology base adjustments to correct the population size of seabirds breeding in difficult to access areas: the case of Cape petrels. If you want to include the distribution in the title, I recommend you take into account the adjusted number of nests in the distribution map and respectively discussion, i.e., instead of only mapping the location of breeding areas you could also represent the number of nests estimated for each of the areas if you have this information by breeding area. However, if your main objective is to show the general lack of changes in the population trend and distribution of the species, then you could change the title to: Contemporaneous and historical (similar) population size and distribution of Cape petrel breeding in the Vestfold Islands, East Antarctica.

Reply: We appreciate the suggestions and have changed the title to "Phenology-based adjustments improve population estimates of Antarctic breeding seabirds: the case of Cape petrels in East Antarctica.

ABSTRACT Although I understand the word limits of an abstract, I would recommend some restructuration of the abstract: For example, begin to explain that it is crucial to know the distribution and population size of a species for conservation and monitorization purposes. Then, explain that some species breed in areas where researchers can only access under specific environmental conditions which may not match with the optimal period to perform population surveys as the number of breeding pairs may change throughout the breeding season. This is the example of Cape petrel breeding in Vestfold Islands in East Antarctica where sea and ice condition limit population surveys. Subsequently explain your main aim and what you have done to achieve it. Delete the sentence from line 14 to 16 as I believe it is not needed in the abstract. Organize the description and discussion of your results an then present a general conclusion of your article. Right now you have for example in lines 18-20 a sentence explaining your results but the previous sentence sounds more like a conclusion and so it should be at end of the abstract

Reply: we have restructured the abstract as suggested by the reviewer, and clearly stated the aim of the study.

KEYWORDS: *The common name of the study species is already in the title and the abstract, instead, you could put its scientific name. Also consider changing the "East Antarctica" keyword that is already in the title and abstract, with other keywords that are also relevant but not included in the abstract, for example, some of the software you use, or remote camera*

Reply: Great suggestion. Words from the title have been deleted and other suggestions by the reviewer included.

INTRODUCTION: *I would also recommend some re-structuring of the introduction. But before any of that, for me, it seems that the objectives of this article are not well defined, although in the title you gave the main relevance to the use of the phenology adjustment approach, at end of the introduction you don't even mention it. So for you, what are your main objectives in this article? Think about it and restructure the introduction accordingly to that. Right now you begin to explain the importance of seabirds as indicators of environmental changes and the difficulties of monitoring them. Afterwards, you refer to some of the methods used to estimate seabird populations (although part of the approaches is mainly referred to those species who nest at ground surface), the difficulty to access remote areas and the need to take into account the variation of the number of active nests throughout the breeding period. You already finished the second paragraph explaining that in this article you used the phenology data from a remote photo-trap camera to apply to your data without explaining which species, why that species, etc. You should present this information later on in the introduction.*

In the third paragraph, you describe several conservation management details about the Antarctic territory but are this needed in the introduction to understand the objective of this article? If you considered that is relevant leave it like it is, otherwise consider putting this information in the methodology section.

You finish this third paragraph with an aim of this article however usually the objectives are presented at the end of the introduction and as it is written here it does not have much sense. What is that you want to inform about CEMP? The population estimation? The population changes from 1974 to 2017? Delete this sentence. In the previous paragraph, you explained why you choose Cape petrels as your study model and here in the fourth paragraph, you describe the species. Although the line of thoughts if correct, some of the information that you put in this paragraph is not relevant for the introduction or its relevance need to be better explained. For example, why is it important their main type of prey? Or the colour of their plumage? If you explain that the colour of their plumage may preclude the use of some methodologies, or that the type of prey is what makes this species a good indicator of the marine environment, then this information would be useful, but right now it does not seem relevant and I recommend you put in the methods.

In the paragraph that begins in line 83, you talk about the environmental characteristics of Antarctica. Consider integrating this information into the third paragraph that you already talk about Antarctica. It is not clear why you mention here the Adele penguin nor why you expect Cape petrels population to be stable instead of increase as in Adele penguins and if there is any study that shows how the wind may influence the breeding distribution of this species or a related species please include it here.

Reply: The review has provided useful feedback on the introduction to better present information to the reader and help articulate our objectives. We have followed the suggestions to restructure paragraphs and have moved some information to the methods under a heading 'Study species and region'.

Specific comments:

Line 29: add " about the effectiveness of" after the word "inform"

Reply: Changed as per above suggestion, Line 29

Lines 61-70: Most of the information in this paragraph should go to methods. Why is it relevant to know their prey in the introduction?

Reply: Retained the key point in the introduction and moved other information to the methods

Line 97: What do you mean by population status? If you are referring to the population trend, what is the difference between objective 1 and 2? Because for you to understand the trend of a population, you need to compare the changes between the population size in at least two different periods and this is what you are already referring to in objective 2.

Reply: We have re-written the aims to clarify our objectives and changed the wording from trend to estimate.

METHODS: Consider to include a "study species" section to put some of the information you included in the introduction but that would fit better here.

Reply: The section in the introduction from lines 61-70 has been moved to the methods under a section titled "Study species and region"

The population surveys section is confusing. Maybe you could organize it per year, i.e., in one paragraph you may explain everything regarding the year 1972 and 1974. Another one to 2017, etc.

Reply: We adopted the suggestion and now have a paragraph explaining the method in each year.

Line 125: It is not clear how the data of 2016 was used

Reply: There is now a dedicated paragraph to explain the 2016/17 survey and how it was used, Line 142-150.

Lines 127-128: I would include here the detailed breeding phenology that you put in lines 64- 66 of the introduction.

Reply: We have moved the breeding phenology information from the introduction into the methods under the heading 'Study species and region'

Line 130: one example is enough, but if you want to put all, include also the one of 2016/17

Reply: We have removed all examples except one, Line 89-90

Line 147: Three? why are you excluding 2016? Is it because only presence-absence was register and not the number of nests?

Reply: The application of the 2016/17 survey is now explained in the dedicated paragraph, Lines 142-150

Lines 141-51: And for those islands that were surveyed in both years did you use only the data of 1974, or did you calculate the mean number of nests for both years?

Reply: We have added detail to the method to clarify the precision of the data in the different survey years. Then outline which data were used and why to estimate the historical population size. Lines 127-141

Line 170: I don't understand why you are referring to this here "were considered consistent with the breeding pairs estimates in historic surveys". Isn't this one of the things you want to verify in this article?

Reply: The intention was to outline how we aligned the counting methods in the 2017/18 ('confirmed nests') and 1970s ('occupied nests') surveys. We have revised this to simply state that both surveys counted 'occupied nests'. Lines 137 and Line 155

Line 184: How was this 10% calculated?

Reply: We have elaborated on the reason for applying the 10% margin of error, and added a reference to the paper that recommended this approach. Lines 202-205

Lines 194-196: But if you don't know when the survey took place how are you going to adjust the survey counts?

Reply: We have elaborated on this ICESCAPE aspect of the analysis, with a section clarifying how the uncertainty in survey dates was accounted for. Lines 211-237

Lines 201-203: It is not necessary to put the sign "-" as you already included S, but you have to put E in the longitude. Include also the dates that the camera stayed in the field collecting data.

Addressed: removed "-" and added E

Lines 201-203: Since you adjust the number of nests based on the pictures of this single-camera I think you need to describe better all the procedures related to it. How did you program the camera? I mean how many pictures per day? How did you choose the place to deploy it? Did you make any population survey to the area covered by the camera to validate the number of nests estimated by the SPPYCAMS software?

Reply: Additional detail of the camera programming and position has been added to the methods. Lines 176-180

Line 2014: I am assuming that in reference 38 it's explaining, but how much is the efficiency of this software to correctly identify the number of occupied nests? And does this software distinguish between breeders and non-breeders (floater or loafing individuals that you refer to in line 172) that also visit the colony?

Reply: Additional detailed of the SPPYCAMS image processing has been added to the methods, specifying that it is manual (not automated process), how many images were counted, and what was counted in the images. Lines 1821-191

Line 206: How did you calculate this repeatability?

Reply: This repeatability refers to the 'count uncertainty' defined earlier in the methods. We have added more detail (including a step by step process) about how the survey data were adjusted and used more consistent terminology to improve clarity. Lines 216-218

Line 234: why is 2016 not included here if you only take into account presence-absence?

Reply: The purpose of the 2016 survey and how the data were used is more clearly articulated in a dedicated paragraph under the 'Population surveys' section and with a statement in the 'Contemporary and historical count data' section of the methods; "the 2016/17 survey was used as a basis to develop count methodology and confirm broad presence-absence prior to 2017/18 survey effort, but did not contribute to the contemporary population estimate or distribution change" Lines 142-150 and Lines 202-205

Line 236-238: If possible, specify which areas were visited and on which dates.

Reply: We added a dedicated paragraph to the "Population surveys" section to better explain the opportunistic observations. Lines 160-164

RESULTS:

Lines 242-243: rephrase maybe to "...November, when no adults were observed"

Reply: Changed as per above recommendation. Line 270

Line 247: Cite figure 2 here. Is this species synchronous? I think this is an important detail because if your species is not synchronous, its phenology may change depending on the environmental conditions and you would be introducing a biased in your estimations.

Reply: Reference to figure 2 added. Line 277. We also added detail to the 'Study species and region' section of the methods explaining that their egg laying is synchronised year to year and, from the little knowledge available, it is synchronised across broad regions of Antarctica. Line 88-90

Line 256: How did you test it?

Reply: Wording changed as a test is not applicable in this instance. "The adjusted nest counts for the Cape petrel population in the Vestfold Islands are similar for the early 1970s to 2017 (Figure 3) and the 95% confidence interval of differences included zero, indicating there was no detectable difference between them." Line 289-293

Line 261: 5 to 763

Reply: Changed as per above recommendation. Line 288

Table 1: To be consistent in the adjusted data columns you should put always "No data" if the number of nests counted was zero as you did for the Northern island or change in Northern islands from "no data" to 0

Reply: Table and caption revised. "Zero indicates the island was searched and no occupied nests were recorded."

Table 1: Why did you put ranges in line 11 y 14, but not in line 8 or 10 of 2017? In lines 8 and 10 were the nests only counted once?

Reply: Table has been revised to include ranges where they were missing

*Table 1 (line 21): What does this asterisk * mean?*

Reply: Typo, removed

Figure 1: maybe you could represent the estimated number of nest per year in the maps, in this way it would be possible to observe geographic changes in the number of nests

Reply: We added the island numbers (as presented in Table 1) so the count and distribution data are more transferable. Given that surveys were not always conducted at the peak nesting period, we feel that the polygons in Figure 1 may not represent the spatial area occupied by nests estimated from the adjusted counts.

Figure 2: Which family did you use in the gam model? And which response variables did you include? Is this the output of the software ICESCAPE?

Reply: We have included additional text about the GAM model in the methods (Lines 215-241) and updated the figure 2 axis labels with consistent terminology to represent the variables used and outputs from the ICESCAPE model.

Figure 3: This information is already in the last line of table 1, so I am not sure which is the value of this figure?

Reply: Yes, you are right that the information is available in the table however, we believe this figure is important as it provides a visual representation for the main point of the study.

DISCUSSION

Lines 353-354: You refer that environmental conditions such as snow accumulation influence the breeding success of the species. Were the environmental conditions between 1974, 2017 and 2019 similar? You should check it and if they were not the same you should discuss possible biases. Furthermore, I would recommend you to continue with the remote camera in the field for several years (if that is possible) to understand how the environmental conditions may affect the number of active breeding pairs and potential variation in the breeding phenology of the species.

Reply: We have simplified the discussion of environmental conditions to focus on how they can influence nesting distribution. Lines 375-387. We also included a statement in the methods to clarify that the camera has remained insitu for ongoing monitoring (Lines 179-180) and detail in the discussion on how this ongoing monitoring has been used to improve this adjustment approach for Adelie penguins, and will enhance this application to Cape petrels. Lines 401-405

Line 399: Not sure why this would implicate a novelty. Both species breed on the ground independently of being walking or flying species

Reply: Deleted the sentence

Line 401: Drones have some advantages but also some limitation. Usually, you can only operate in specific environmental conditions (low wind speed), the surveys are limited by the duration of the battery, it may scare the birds which may fly away, there can be a limitation regarding the maximum distance between the operator and the drone and this may limit its use depending on the environmental conditions. Satellite images have also been used to count penguins, boobies or albatrosses nests, but not sure if this would be feasible with Cape Petrels due to a lack of contrast between the bird's plumage and the ground and because of their small body size

Reply: Points noted. We have changed the reference of 'drones' to 'aerial photography' and moved this information to a dedicated 'limitations' paragraph of the discussion. Lines 406-412

Reviewer: 2

Comments to the Author(s)

This study aims at showing that one Cape petrel population in East Antarctica has been stable in the last ca. 50 years. Authors used survey data adjusted for seasonal changes in the number of breeders. While the study raises an interesting and important issue, I think it fails to convincingly show that the

*Cape petrel population has been stable. The study is based on more or less two surveys (1974 and 2017) and uses data from another season (2019) to make some phenological adjustment. The conclusions **assume that the phenological changes in the number of breeders during the 2019 season are the same as in 1974 and 2017. This is a very strong assumption and authors do not provide any evidence that this is the case.** Consequently, there is no evidence that the adjusted counts given by the authors in this study are better than the raw counts and the trend in this petrel population cannot be assessed with certainty.*

Reply: We have adjusted the language in the paper to remove statements about population trends and focused on the methods and providing a population estimate to base future monitoring. As suggested by reviewer 1, we have added detail about the spatio-temporal variability in Cape petrel phenology (methods Lines 86-91) and a limitations paragraph in the discussion to elaborate on the implications of this knowledge gap for interpreting our results (Lines 388-405)

Additional comments:

Line 35-36: just a detail but photographic ground surveys and surveys from remotely operated cameras are also ground counts. Consider changing "ground count" line 35 with "direct count" or equivalent.

Reply: Our revision of the introduction (as per Reviewer 1 suggestions) removed reference to ground/direct counts and now refers to population/survey counts.

Line 48-51: there are no national territories in Antarctica, only claims. You should modify this section as such statements are not necessary in a scientific paper and only open the door for criticism. You should only mention East Antarctic and the role of CCAMLR without getting into such administrative/political considerations.

Reply: Modified as suggested, to focus on East Antarctica and the role of CCAMLR.

*Line 58: you could specify Antarctic krill *Euphausia superba* here. Also, should it be "the Cape petrel is an indicator species..." rather than "cape petrels are an indicator species..."?*

Reply: Changed as per above recommendation. Line 42-43

Line 85: the SAM is not a local environmental parameter and not specific to the Prydz Bay. This is a large climatic mode not relevant here to discuss local environmental changes.

Reply: We have removed the reference to SAM.

Lines 87-88: mentioning changes in sea-ice here while you just said that the environment, including sea-ice was stable, is a bit strange?

Reply: Added 'Seasonal changes' to distinguish from change associated with long-term climate. Line 59

Lines 83-95: this paragraph is a bit blurry. You mention a stable environment but then some environmental changes and then put the Adelie penguin into the story... I don't think this is needed into your story to argue that the environment has been stable or not (at least not at this stage in the introduction) so I would suggest to remove this paragraph. If you decide to keep it, you should present some more solid arguments to conclude that the foraging habitat of the Cape petrel has been stable since 1958.

Reply: The paragraph has been revised. We removed reference to Adelie penguins, distinguish seasonal from longer-term change, and emphasis "In the context of these mixed

signals of environmental change in Prydz Bay and East Antarctica, it is useful to assess if the population size and distribution of Cape petrels in the Vestfold Hills has remained stable or changed in recent decades.” Line 62-65

Methods:

Population surveys: this is not so clear to me why 2016 is not included in the analyses? The 2016 survey is mentioned line 137-139 but then does not appear anywhere? 1972 is barely used as well... overall, I find it hard to follow the survey descriptions and what has been used or not... Timing, spatial coverage, methods.... seem to vary among survey years so that any comparison between these numbers are based on very soft grounds, whatever the adjustments you make.

Reply: We followed the suggestion from reviewer 1 and now have dedicated paragraphs to articulate survey methods and application for each year. Then state “the 2016 survey was used as a basis to develop count methodology and confirm broad presence-absence prior to 2017 survey effort, but did not contribute to the contemporary population estimate or distribution change.” Line 199-201

Line 183-185: where does this come from? Is this based on any data or is this just a guess?

Reply: As per reviewer 1 suggestion, we have clarified the ‘count uncertainty’ and reference a paper for context. Line 202-205

Lines 198-201: then you assume that there is no change in this egg laying dynamics among years?... this is a very strong, and likely unrealistic, assumption

Reply: This was raised by reviewer 1 as well, and address in the Study species and region section of the methods and a limitations paragraph of the discussion.

Line 206-207: this is really misleading to call “repeatability” the 10% arbitrary threshold used to define potential variation around the count estimate. Repeatability has a specific definition which does not correspond to yours here.

Reply: This has been revised (previous comment) and we have elaborated on the adjustment method using ICESCAPE, including a step-by-step process. Line 216-237

Discussion

Line 315: hard to conclude with 2 datapoints...there could have been a lot of changes in this period...

Reply: Agreed. We have revised the wording in the aim of the paper, throughout the results and here in the discussion.

Line 337-338: high site fidelity and stability are two different things. You may have a stable population even if site fidelity is low and vice versa. This section is mixing both concepts.

Reply: Agreed. This paragraph has been revised and now focuses on population estimates. Line 356-374. The distribution change is then discussed in the next paragraph Line 375-387.

Line 368: “changes in sea-ice conditions” instead, not “sea ice conditions” per se

Reply: Changed to ‘sea-ice’ throughout the manuscript.

Line 399-400: but this is doable for any kind of organism. The key is not to do it, but to show whether or not it gives accurate numbers. Your study has used this “adjustment” method and made conclusions based on it but the validity of this method is still unknown for your system.

Reply: We deleted this sentence on advice from both reviewers and added a limitations paragraph to the discussion that addresses this concern. Line 388-412

Appendix C

Response to reviewers:

Associate Editor (Dr Denise Greig)

Comments to Author:

Thank you so much for your thoughtful revision and response to the reviewers. I think the manuscript is in good shape. There are just a few additional requests from the reviewers and three typos that I noted:

Line 115. Change “focussed” to “focused”

Reply: change to ‘focused’

Line 243. Is there supposed to be a “.” after “100”?

Reply: decimal removed

Line 244. Move “Temporal population change” to its own line.

Reply: moved to new line

Reviewer comments to Author:

Reviewer: 1

Comments to the Author(s)

The authors did a great job improving the manuscript and I believe they addressed all the comments pointed out by the previous review. I have just a few minor comments (mostly typos):

Abstract

Line 13: add a comma after “In this study”

Reply: comma added

Line 14: I know that it can be complicated due to the usual word limitation of the abstract, but I think it would be important to refer that you used photographs obtained from a remote camera to adjust the nest counts.

Reply: added “using photographs from remote cameras”

Introduction

It is better structured now and the aims are well defined.

Line 37: I would add commas before and after “such as Antarctica”

Reply: commas added

Line 38: I would add commas before “such as sea ice”

Reply: commas added

Line 54: It would be better to include references based on Cape petrels studies and not only include a review of seabird threats around the world and from different seabird species. Some references could be:

Favero, M., Khatchikian, C.E., Arias, A., Silva Rodriguez, M.P., Cañete, G. and Mariano-Jelicich, R. (2003). Estimates of seabird by-catch along the Patagonian Shelf by Argentine longline fishing vessels, 1999–211106111061. *Bird Conservation International*. 13(4): 273–281.

Braun, C., Esefeld, J., Savelieva, L., & Peter, H. U. (2021). Population decline of the cape petrel (*Daption capense*) on King George Island, South Shetland Islands, Antarctica. *Polar Biology*, 44(9), 1795-1801.

Reply: references added

Methods

Line 94: something is missing after the scientific name of the silverfish, maybe add “being” before “an addition...”

Reply: now reads ‘...silverfish *Pleuragramma antarcticum* being an addition...’

Line 96: Add “of Nature” after Conservation

Reply: added ‘of Nature’

Line 143: Could the petrels be on pre-laying exodus during this survey?

Reply: Added ‘; this was after birds have returned to breeding sites and prior to the pre-laying exodus {Johnstone, 1973 #67}.’

Line 242: I believe this is point 5

Reply: added point 5

Line 244: Put the title of the subsection in a new line

Reply: moved title to new line

Results

Line 274: Figure 2 does not seem to reach the 8th of March

Reply: That is correct. We stopped counting camera images earlier than this because the data were not required to adjust population surveys

Line 280-282: This is not a result, I suggest you move it to the discussion

Reply: removed

Line 288-291: I believe it will be better to describe first the historical data and then the contemporaneous, but this is just a suggestion

Reply: The order was deliberate to fit with our aims of getting a population estimate and then comparing with historical surveys

Line 290: I believe you meant 16 and not 13 islands, based on table 1

Reply: 13 islands is correct. Added 'In 2017/18,' to the start of the sentence to clarify the 13 relates to the 2017/18 survey.

Line 295 – 296: This is not a result, I suggest you move it to the discussion

Reply: removed

Discussion

Line 350: add a comma after "Here"

Reply: added comma

Line 352: it would be good if you point out somewhere in the discussion that this methodology is useful for surface breeders, as many other flying seabirds breed in burrows and your methodology probably would not be feasible or at least not easily applicable

Reply: changed 'flying seabirds' to 'surface-nesting, flying seabirds'

Line 389: Please put Cape with a capital letter.

Reply: Changed to capital C

Line 411: I don't understand what you mean by 'the body of nest attendance data'

Reply: Change 'body' to 'amount'

Reviewer: 3

Comments to the Author(s)

Based on the comments from the previous reviews, I think the authors have done a good job in improving the manuscript. I found the methodology clearly set out. However, it is still a limitation that you do not ground truth your results, and you claim (now in your title) that this approach improves estimates its surely weakened by this fact. You apply the same approach to two years and

assume that the results you get are more accurate.

While some specific aims of the paper are set out in the abstract and end of the introduction, the overall aim of the paper is still not that clear. Is it a proof of concept paper? Or a study to determine changes in the population trend of Cape petrels? One sentence on this should be explicitly written in the abstract and the end of the introduction.

If this is proof of the use of combining phenological data with survey data to estimate counts then I was surprised by the lack of review of literature in the introduction, on existing methodological approaches to do so. Would be nice for the reader to know how developed this is as an approach. Despite these points I do think this is an interesting and valuable paper, highlighting an approach which could be highly valuable for survey species in challenging environments. I found it well written and structured, especially after the comments from previous reviewers.

Some very minor comments:

I. 53-56 would move these sentences about threats to petrels a bit earlier

Reply: moved to second sentence of the paragraph

I 88 – is this an appropriate reference?

Reply: added Johnstone 1973, in addition to Pinder 1966

fig 1 change order in label so its yellow (1970s) blue (1970s and 2017/18) and then orange (2017/18)

Reply: change to suggested order

116-119 why not historical vs contemporary?

Reply: I suspect this refers to line 257-259, but I'm not sure. We prefer to reference the contemporary survey first then the historical (throughout the ms), as this aligns better with the aims of the study.

I. 264 'R framework?' More conventional to refer to a package

Reply: No R package was used to digitise the maps. I have changed to 'R v4.1.1'